# PD-1 suppresses TCR-CD8 cooperativity during T-cell antigen recognition

Kaitao Li [1,2], Zhou Yuan[2,3], Jintian Lyu[1,2], Eunseon Ahn[4,5], Simon J. Davis[6], Rafi Ahmed [4,5] & Cheng Zhu [1,2,3✉]

Despite the clinical success of blocking its interactions, how PD-1 inhibits T-cell activation is incompletely understood, as exemplified by its potency far exceeding what might be predicted from its affinity for PD-1 ligand-1 (PD-L1). This may be partially attributed to PD-1's targeting the proximal signaling of the T-cell receptor (TCR) and co-stimulatory receptor CD28 via activating Src homology region 2 domain-containing phosphatases (SHPs). Here, we report PD-1 signaling regulates the initial TCR antigen recognition manifested in a smaller spreading area, fewer molecular bonds formed, and shorter bond lifetime of T cell interaction with peptide-major histocompatibility complex (pMHC) in the presence than absence of PD-L1 in a manner dependent on SHPs and Leukocyte C-terminal Src kinase. Our results identify a PD-1 inhibitory mechanism that disrupts the cooperative TCR–pMHC–CD8 trimolecular interaction, which prevents CD8 from augmenting antigen recognition, explaining PD-1's potent inhibitory function and its value as a target for clinical intervention.

[1] Wallace H. Coulter Department of Biomedical Engineering, Georgia Institute of Technology, Atlanta, GA, USA. [2] Parker H. Petit Institute for Bioengineering and Biosciences, Georgia Institute of Technology, Atlanta, GA, USA. [3] George W. Woodruff School of Mechanical Engineering, Georgia Institute of Technology, Atlanta, GA, USA. [4] Emory Vaccine Center, Emory University School of Medicine, Atlanta, GA, USA. [5] Department of Microbiology and Immunology, Emory University School of Medicine, Atlanta, GA, USA. [6] Radcliffe Department of Medicine and Medical Research Council Human Immunology Unit, University of Oxford, John Radcliffe Hospital, Headington, Oxford, UK. ✉email: cheng.zhu@bme.gatech.edu

D espite the great success of targeting programmed cell death 1 (PD-1) or PD-1 ligand 1 (PD-L1) to modulate T cell functions for immunotherapy[1–3], the mechanisms of how PD-1 suppresses antigen-specific T cell responses are not fully understood. PD-1 signaling is initiated by the phosphorylation of its immunoreceptor tyrosine-based inhibitory motif (ITIM) and immunoreceptor tyrosine-based switch motif (ITSM) when PD-1 and the T-cell receptor (TCR) are co-engaged with their respective ligands. This leads to the recruitment and activation of Src homology region 2 domain-containing phosphatases (SHPs), which attenuates the phosphorylation-dependent signaling cascades downstream of the TCR and the co-stimulatory receptor CD28, and thereby suppresses cellular functions such as activation, proliferation, metabolic regulation, cytotoxicity, and cytokine production[4–8]. Titration of PD-1 expression on T cells demonstrated its potent inhibitory signaling, such that even very low PD-1 expression is able to inhibit TNF-α and IL-2 production as well as T-cell proliferation[9]. The potency exceeds what might be predicted from its ligand-binding affinity, which is 1–2 log lower than that of B7-1 interacting with cytotoxic T-lymphocyte-associated protein 4 (CTLA-4), another inhibitory receptor of the same family[10,11]. Such high potency might be partially attributed to PD-1's inhibition of the early activating signals such as the phosphorylation of CD3ζ and ZAP70[6,8,12], the immediate steps following TCR triggering, and the phosphorylation of CD28[13]. Yet, it remains unclear whether PD-1 could function at even earlier stages to perturb the initial antigen recognition process.

T-cell antigen recognition requires the specific interaction with the cognate peptide-major histocompatibility complex (pMHC) by the TCR, which leads to the phosphorylation of CD3 immunoreceptor tyrosine-based activation motifs (ITAMs) and a variety of proximal signaling molecules[14]. Far more complex than kinetic analysis in fluid phase using purified proteins, which reports primarily the effect of the physiochemical properties of the intermolecular interface, in situ measurement of interactions across the cell-cell junction reveals highly dynamic binding characteristics under force-free or force-loaded conditions[15–23]. Moreover, intercellular interactions are regulated by both cell-intrinsic and -extrinsic factors from the local environment that tune the antigen recognition process[15,16,20]. T-cell antigen recognition is greatly enhanced by the engagement of the co-receptor CD8 with pMHC, which stabilizes the TCR–pMHC complex and keeps the Leukocyte C-terminal Src kinase (Lck) in the proximity of TCR-CD3. Underlying this enhancement is the TCR-signaling-induced upregulation of CD8 binding and formation of cooperative TCR–pMHC–CD8 trimolecular bonds, most likely via the recruitment of CD8 to TCR-CD3[17,24–26] through a Lck bridge[27–31]. This "inside-out" signaling augmentation suggests an additional potential mechanism to regulate T-cell antigen recognition.

In this study, we analyzed the effect of PD-1 on T-cell antigen recognition, utilizing P14 TCR-transgenic CD8+ T cells and the cognate antigenic peptide Lymphocytic Choriomeningitis Virus (LCMV) gp33-41 bound to H2-Db MHC. We observed reduced pMHC-mediated cell spreading when co-engaging PD-1 with PD-L1 despite more ligands being presented on the surface, suggesting PD-1 suppression of T-cell interacting with its antigen. In situ kinetic analysis of T cell interactions with pMHC, PD-L1, or both separately and concurrently further demonstrated fewer numbers and shorter lifetimes of bonds formed concurrently with both ligands than the sum of bonds formed separately with each, revealing "negative cooperativity", a phenomenon opposite to synergy. Furthermore, the negative cooperativity depended on SHP and Lck activities, extracellular pMHC–CD8 binding, and intracellular CD8–Lck association, suggesting that PD-1 primarily disrupts the positive cooperativity between TCR and CD8 during

antigen recognition. Moreover, the negative cooperativity impaired CD8 augmentation of downstream Ca²⁺ signaling. These data reveal a mechanism in which PD-1 fine-tunes antigen recognition via "inside-out" negative feedback.

## Results

**PD-1 suppresses T cell spreading on pMHC surface**. We first examined T cell spreading on coverslips functionalized by pMHC with or without PD-L1 (Fig. 1). CD8+ T cells from $PDCD1^{-/-}$ P14 TCR-transgenic mice re-expressing PD-1 or vehicle spread on gp33:H2-Db (but not BSA) surface (Fig. 1a–c), indicating that the antigen recognition machinery alone is able to support spreading without adhesion molecules (e.g. integrins) used in most immunological synapse (IS) studies[32]. Spreading area on PD-L1 was much smaller, but above background as controlled using BSA surface or T cells expressing vehicle (Fig. 1a–c). Although each ligand (gp33:H2-Db or PD-L1) alone was able to mediate cell spreading individually, when both ligands was co-presented the spreading area was greatly reduced for PD-1-expressing cells but unchanged for vehicle-expressing cells (Fig. 1a–c). Despite that the total ligand density in the co-presentation case was the sum of those in individual coating, the cell spreading area was much smaller than that on gp33:H2-Db alone. This effect was further confirmed for activated CD8+ T cells from wild-type (WT) P14 mice, which expresses endogenous PD-1 (Fig. 1d, e). In contrast, spreading areas on surfaces co-presenting gp33:H2-Db with Intercellular Adhesion Molecule-1 (ICAM-1) were larger than those on surfaces presenting gp33:H2-Db or ICAM-1 alone (Fig. 1f, g), consistent with the previous reports of ICAM-1 enhancement of Jurkat cell spreading on anti-CD3 surface[33] and of the positive cooperativity between TCR and LFA-1 in binding to their respective ligands[34]. Together these data suggest that PD-1 suppresses the antigen recognition process that relies on binding of the TCR-CD8 axis to the cognate pMHC, an opposite effect to LFA-1 enhancement of IS formation.

**PD-1 reduces T cell bond formation with pMHC**. To define the mechanisms causing T cells to spread less on pMHC and PD-L1 than on pMHC alone, we analyzed two-dimensional (2D) interactions between a T cell and pMHC, PD-L1, or both presented by a human red blood cell (RBC) as a surrogate antigen-presenting cell (APC) using the adhesion frequency assay[11,35,36]. An activated P14 T cell was aspirated by a micropipette (Fig. 2a, right) to interact in random sequence with three RBCs coated with known densities of gp33:H2-Db, PD-L1, or both, aspirated by another micropipette (Fig. 2a, left). For each RBC, the T cell was driven by a piezoelectric motor to make 5-s contact cycles repeatedly. The outcome of each cycle is a binary score: 1 or 0 for adhesion or no adhesion (as detected by the presence or absence of RBC elongation when the two cells were separated). An adhesion frequency ($P_a$ = average adhesion scores) was obtained for each RBC after 30–50 cycles of T cell contacts and three $P_a$ values were obtained for each T cell (Fig. 2a, b). The readout is highly specific, as coating streptavidin (SA) alone or replacing the cognate peptide gp33 with gp276 on H2-Db abolished RBC binding to P14 T cells (Supplementary Fig. 1). We calculated the average number of bonds per contact $<n>$ from each $P_a$ ($= 1 - e^{-<n>}$) using a logarithmic transformation (Fig. 2a) because for adhesions mediated by dual species molecular interactions, the $<n>$, but not $P_a$, values of the two species are additive, provided that the two ligand species interact with their respective receptors independently rather than cooperatively[37–39]. As such, cooperativity of T cell binding to pMHC and PD-L1 could be detected by comparing the bond number measured from the dual-ligand RBC to the sum of bond numbers measured from the two single-ligand

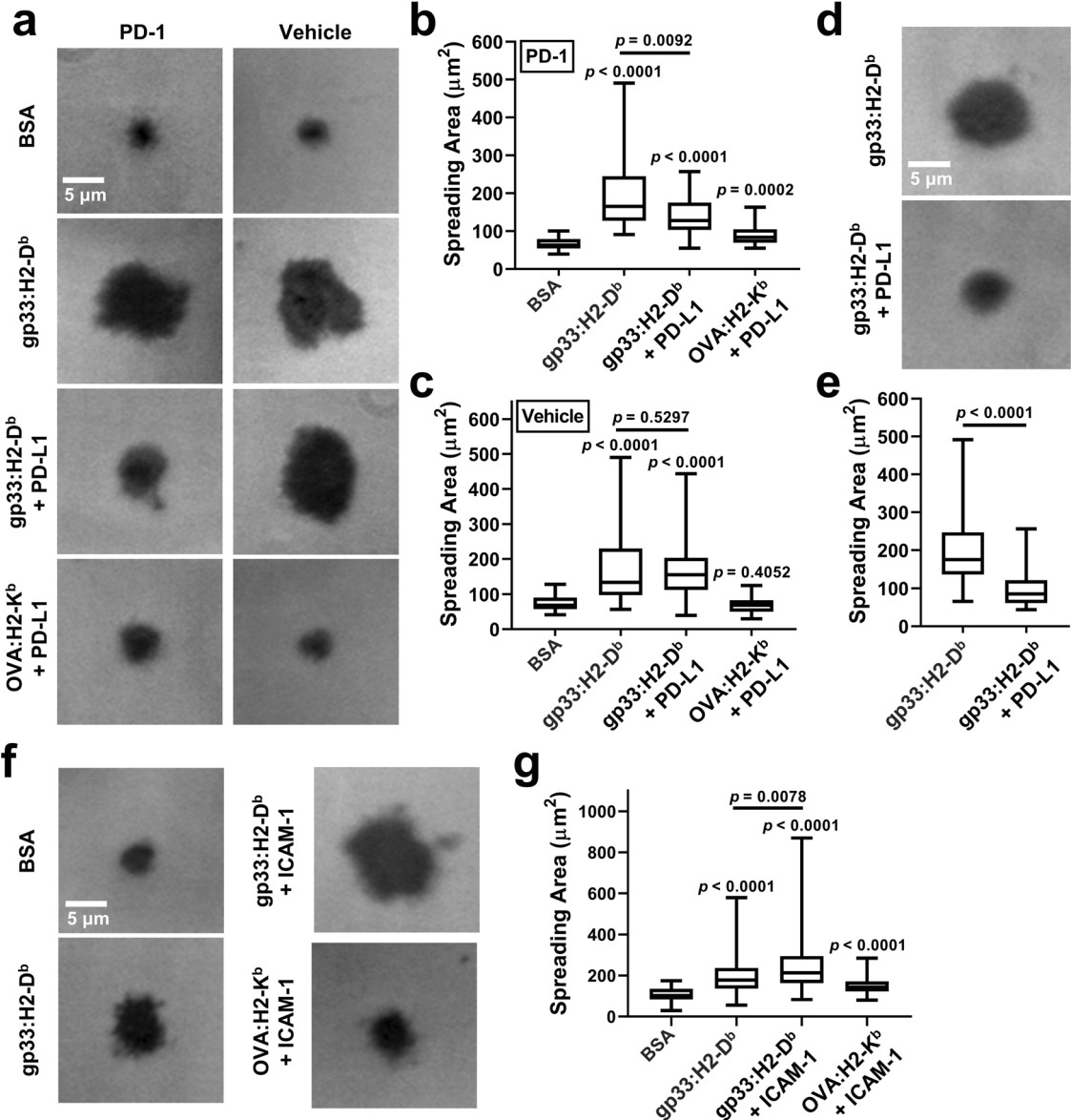

**Fig. 1 PD-1 inhibits T-cell spreading on surface co-presenting pMHC and PD-L1. a** Representative images by reflection interference contrast microscopy (RICM). *PDCD1* $^{-/-}$ P14 transgenic CD8$^+$ T cells re-expressing PD-1 (left column) or vehicle (right column) were added onto glass coverslip coated with BSA (1st row), gp33:H2-D$^b$ (2nd row), gp33:H2-D$^b$ and PD-L1 (3rd row), or OVA:H2-K$^b$ and PD-L1 (4th row). Cell spreading was imaged by RICM 20 min after their addition. **b** Quantification of **a** showing reduced spreading of PD-1-expressing cells on surfaces co-presenting gp33:H2-D$^b$ and PD-L1 ($n = 31, 31, 32,$ and 31 cells). **c** Quantification of **a** showing no change in spreading area of vehicle-expressing cells on surfaces co-presenting gp33:H2-D$^b$ and PD-L1 ($n = 28, 37, 68,$ and 27 cells). **d** Representative RICM images showing the spreading of in vitro activated wild-type P14 transgenic CD8$^+$ T cells expressing endogenous PD-1 20 min after adding onto glass coverslip coated with gp33:H2-D$^b$ (top) or gp33:H2-D$^b$ and PD-L1 (bottom). **e** Quantification of **d** showing reduced spreading of endogenous PD-1-expressing cells on surfaces co-presenting gp33:H2-D$^b$ and PD-L1 ($n = 56$ and 56 cells). **f** Representative RICM images showing the spreading of in vitro activated wild-type P14 transgenic CD8$^+$ T cells 20 min after adding onto glass coverslip coated with BSA, gp33:H2-D$^b$, gp33:H2-D$^b$ and ICAM-1, or OVA:H2-K$^b$ and ICAM-1 (indicated). **g** Quantification of **f** showing larger spreading area on surfaces co-presenting gp33:H2-D$^b$ and ICAM-1 than on gp33:H2-D$^b$ or ICAM-1 surfaces alone ($n = 62, 83, 61,$ and 56 cells). Data are presented by box-whisker plots with the center line labels median, the box contains the two middle quantiles, and the whiskers mark the min and the max. *p* values were calculated using Mann–Whitney test by comparing each group with BSA control or two groups as labeled.

RBCs after matching the ligand site densities (Fig. 2a, c). Strikingly, the measured dual-species bond number was significantly smaller than the sum of single-species bond numbers, such that the "whole" was ~20% smaller than the sum of the "parts" ($\Delta<n>/<n>_{pred}$, Fig. 2c, f), resembling the observation in the cell spreading assay. We call this "negative cooperativity" because it is the inverse of the positive cooperativity between TCR and CD8, as well as between TCR and LFA-1, where dual-species bonds are more than the sum of single-species bonds[17,24,34]. Importantly,

shortening the contact time to 0.5 s substantially reduced the adhesion frequency to PD-L1 and abolished the negative cooperativity (Fig. 2d–f), similar to the previously observed elimination of positive cooperativity between TCR and CD8 for pMHC binding[17,24] and between TCR–pMHC binding and LFA-1–ICAM-1 binding by reducing the contact time[34]. This suggests no direct physical interference due to ligand co-presentation, but instead a temporal requirement most likely involving the cross-talk of these two signaling axes. Moreover, PD-1 deficient P14

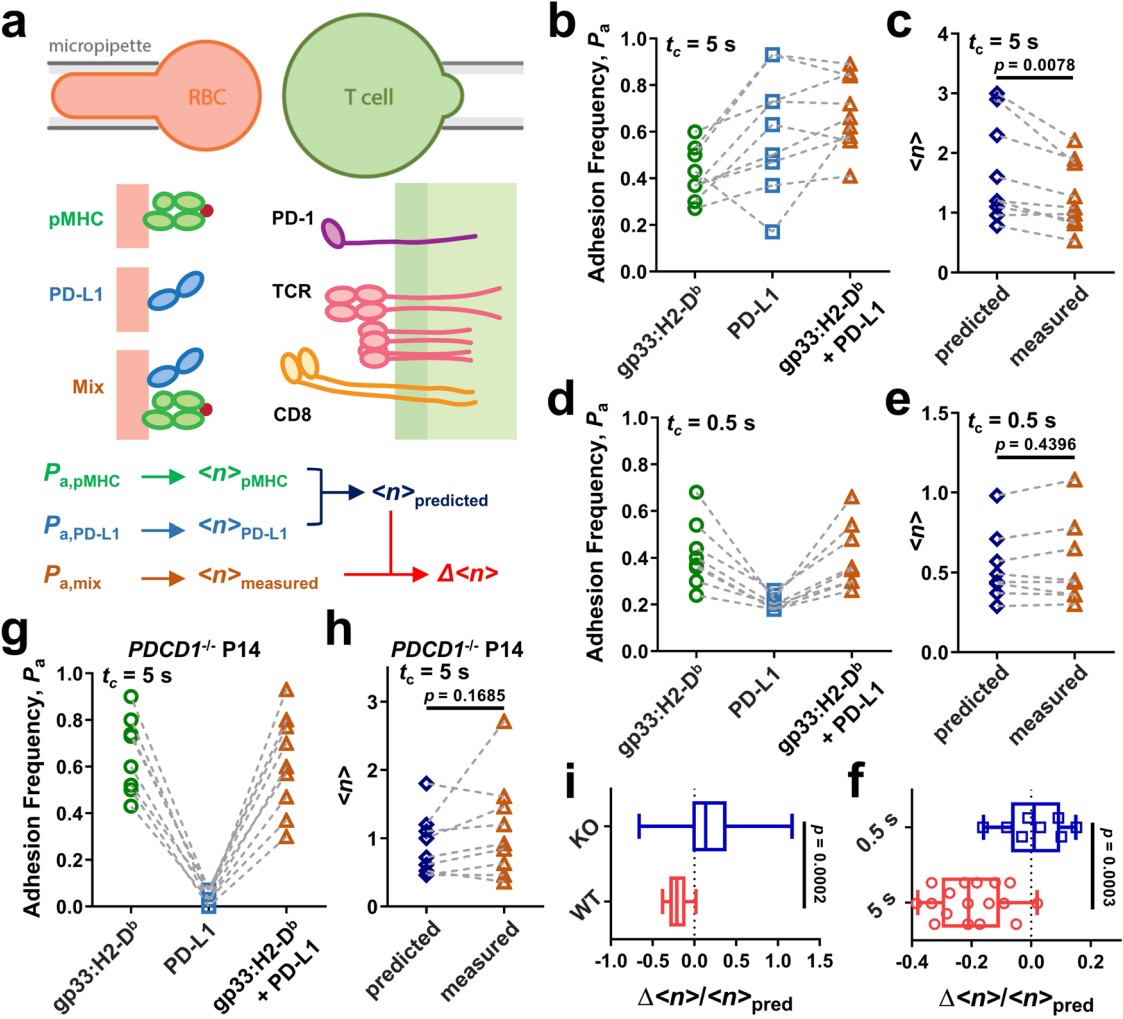

**Fig. 2 2D kinetic analysis of negative cooperativity. a** Schematic of the micropipette assay (top) and the molecules to be analyzed (middle). A T cell expressing TCR, CD8 and PD-1 (right) was tested against three RBCs bearing pMHC, PD-L1, or both (mix) in random order (left) to generate three adhesion frequencies ($P_a$'s), one for each RBC after 30–50 repeated touches with the T cell. The bottom row shows the workflow of using the three $P_a$'s to determine the bond numbers ($<n>$'s) and the differential bond number ($\Delta <n>$). **b, d** Representative $P_a$'s of individual in vitro activated P14 CD8$^+$ T cells binding to RBCs coated with gp33:H2-D$^b$, PD-L1, or both at 5-s (**b**) or 0.5-s (**d**) contact time. Data points measured by testing the same T cell against three RBCs bearing different ligands are connected by a dashed line. **c, e** Comparisons of predicted and measured $<n>$'s from **b** (**c**) and **d** (**e**), respectively. **f** Normalized change of bond number ($\Delta <n>/<n>_{pred}$). The reduction of bond formation observed at 5-s contact tests ($n = 17$ cells) was abolished when the contact time was shortened to 0.5 s ($n = 8$ cells). **g** Representative $P_a$'s of individual activated $PDCD1^{-/-}$ P14 CD8$^+$ T cells to RBCs bearing gp33:H2-D$^b$, PD-L1, or both at 5-s contacts. **h** Comparison of predicted and measured $<n>$'s from **g**. **i** $\Delta <n>/<n>_{pred}$ showing the reduction of bond formation observed when PD-1 expressing cells were used was abolished when $PDCD1^{-/-}$ cells were used ($n = 16$ cells). Data are presented by box-whisker plots with the center line labels median, the box contains the two middle quantiles, and the whiskers mark the min and the max. $p$ values were calculated using paired Student's $t$ test (**c, e, h**) or Mann–Whitney test (**f, i**).

T cells showed negligible adhesion frequencies to PD-L1 RBCs (Fig. 2g) and formed indistinguishable numbers of bonds with RBCs coated with gp33:H2-D$^b$ alone and both gp33:H2-D$^b$ and PD-L1 (Fig. 2h), further confirming that the suppressed binding was mediated by PD-1 (Fig. 2i). We also found similar normalized bond numbers of CD8$^+$ T cells from P14 mice infected with LCMV strain Armstrong (5 days post infection) or Clone 13 (5 or 8 days post infection) to RBCs bearing PD-L1 (Supplementary Fig. 2a) or gp33:H2-D$^b$ (Supplementary Fig. 2b), yet fewer bonds to dual-ligand RBCs than the sum of bonds to single-ligand RBCs for all conditions (Supplementary Fig. 2c), suggesting PD-1-mediated negative cooperativity for in vivo activated T cells during responses to antigen.

**The negative cooperativity depends on signaling of both PD-1 and TCR.** To investigate the intracellular underpinnings of the negative cooperativity involving dual ligand binding to the extracellular domains of PD-1 and TCR-CD8 axis, we perturbed PD-1 signaling by inhibiting SHP1 and SHP2 with NSC87877. No difference was observed in the number of bonds between NSC87877-treated and untreated T cells formed with RBCs bearing PD-L1 (Fig. 3a, normalized by the PD-1 and PD-L1 densities) or gp33:H2-D$^b$ (Fig. 3b, normalized by the TCR and pMHC densities), indicating that the separate binding of T cells to each ligand requires no SHP1 or SHP2 activity. However, the NSC87877-treatment eliminated the negative cooperativity as T cells formed as many bonds with RBCs bearing both ligands as the sum of bonds formed

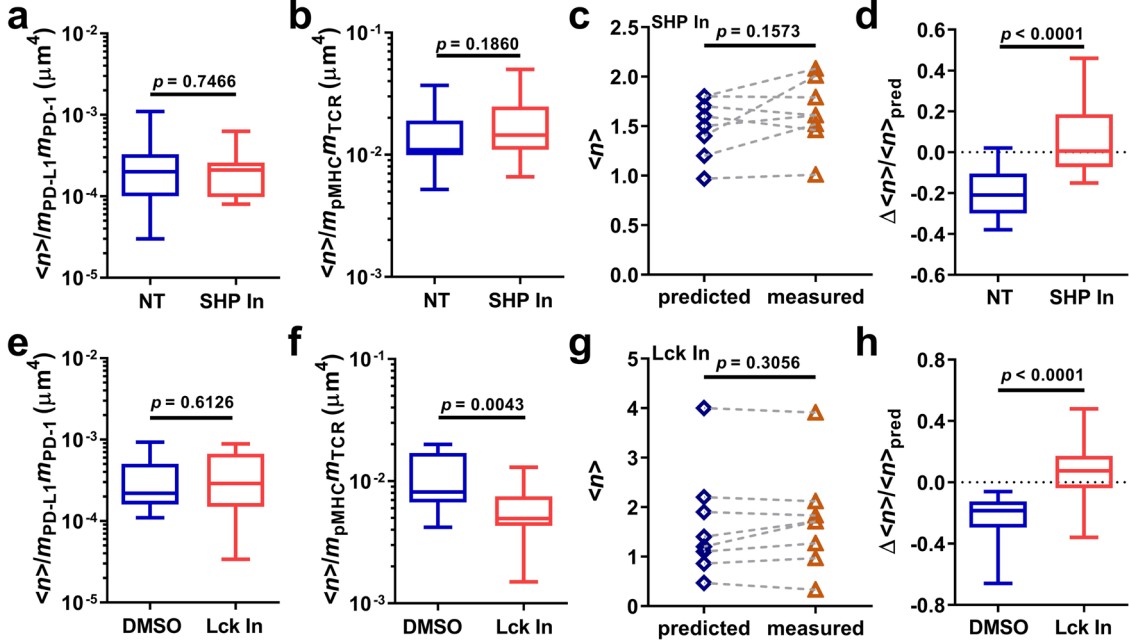

**Fig. 3 Negative cooperativity depends on signaling of both PD-1 and TCR. a, b** Comparisons of normalized numbers of single-ligand bonds formed between PD-L1 (**a**, $n = 42$ and 14 cells) or pMHC (**b**, $n = 27$ and 14 cells) coated RBCs and in vitro activated T cells that were not treated (NT) or treated with 20 μM SHP1 and SHP2 inhibitor NSC87877 (SHP In). **c** Representative results comparing predicted and measured $<n>$'s determined from binding of NSC87822-treated T cells to RBCs bearing pMHC, PD-L1, or both. **d** Comparison of normalized bond reduction ($\Delta<n>/<n>_{pred}$) between the not treated and NSC87877-treated groups ($n = 17$ and 14 cells). **e–h** Similar to **a–d** except that the not treated group was replaced by the DMSO group and NSC87877 was replaced by the Lck inhibitor (2 μM). $n = 14$ and 15 cells in **e**. $n = 14$ and 16 cells in **f**. $n = 14$ and 16 cells in **h**. Data are presented by box-whisker plots with the center line labels median, the box contains the two middle quantiles, and the whiskers mark the min and the max. $p$ values were calculated using paired $t$-test (**c**, **g**) or Mann–Whitney test (the rest).

with RBCs bearing each ligand (Fig. 3c, d), indicating the dependence of negative cooperativity on SHP-mediated PD-1 signaling.

To examine the interplay between activating and inhibitory signals of TCR and PD-1 manifested at the level of ligand-binding, we analyzed cells treated with Lck kinase inhibitor 7-Cyclopentyl-5-(4-phenoxyphenyl)-7H-pyrrolo[2,3-d]pyrimidin-4-ylamine[26,40,41]. As expected, inhibition of Lck had no effect on PD-1–PD-L1 binding (Fig. 3e) but significantly reduced T cell binding to gp33:H2-D$^b$-coated RBCs by ~47% (Fig. 3f), most likely due to the loss of Lck-dependent TCR-CD8 positive cooperativity as we previously reported[17,24]. Inhibiting Lck also abolished the negative cooperativity of T- cell binding to RBCs bearing dual ligand species (Fig. 3g). The requirement of both PD-1 inhibitory signals (Fig. 3d) and TCR activating signals (Fig. 3h) for the negative cooperativity rules out a simple linear model for the dual-ligand binding of T cells. Instead, these data suggest an "inside-out" signaling mechanism, by which con-current ligand binding of PD-1 and the TCR-CD8 axis is regulated by the interplay of the signals they each trigger. It is also consistent with the previous observation that PD-1 phosphorylation and its subsequent SHP2 recruitment can be greatly enhanced by co-engagement of PD-1 and TCR with their respective ligands[8].

**The negative cooperativity requires CD8 binding to pMHC.** Since the readouts of the negative cooperativity – cell spreading and 2D binding – are based on receptor–ligand interactions, it begs the question as to which molecular interaction is suppressed. It seems unlikely that the PD-1–PD-L1 interaction is suppressed since PD-1 ligand-binding was not affected by inhibition of the key phosphatase SHP1 and SHP2 (Fig. 3a) or kinase Lck (Fig. 3e). On the other hand, the synergistic binding of P14 TCR and CD8

to gp33:H2-D$^b$ is sensitive to Lck activity (Fig. 3f). Previous studies from our group and others have demonstrated that cooperative binding of TCR and CD8 to cognate pMHC is a highly dynamic process with positive feedback relying on Lck and the signal triggered by TCR–pMHC binding[17,24–26]. Therefore, the elimination of negative cooperativity by inhibiting Lck could also be explained, at least partially, by the elimination of the target of the negative regulation. To test this possibility, we presented gp33 with H2-D$^b$α3A2, a mutated form of H2-D$^b$ where the mouse α3 domain was replaced with that from human HLA-A2 to abolish CD8 binding while preserving TCR binding[42]. As expected, preventing CD8 binding yielded significantly fewer (~57%) bonds from T cells interacting with RBCs bearing gp33:H2-D$^b$α3A2 than gp33:H2-D$^b$ (Fig. 4a). Supporting our hypothesis, no negative cooperativity was observed when co-presenting T cells with PD-L1 and gp33:H2-D$^b$α3A2 despite normal Lck activity (Fig. 4b, c). These data support our hypothesis that the TCR-CD8 cooperativity is a target of PD-1's inhibitory signal.

**Force-lifetime spectroscopy confirms PD-1 suppression of TCR-CD8 cooperativity.** To further validate our hypotheses regarding the negative cooperativity, we performed force spectroscopic analysis using the biomembrane force probe (BFP) technique by which force-dependent bond lifetimes were measured as interaction characteristics orthogonal to the kinetic parameters measured in the absence of force[18,19,26,43]. Gp33:H2-D$^b$, PD-L1, or both were coated on a glass bead attached to the apex of a micropipette-aspirated RBC (Fig. 4d), which translates bead displacements into force data by multiplying by the cali-brated spring constant of the BFP[36,44], enabling force measurement with sub-piconewton precision over time. The force-clamp

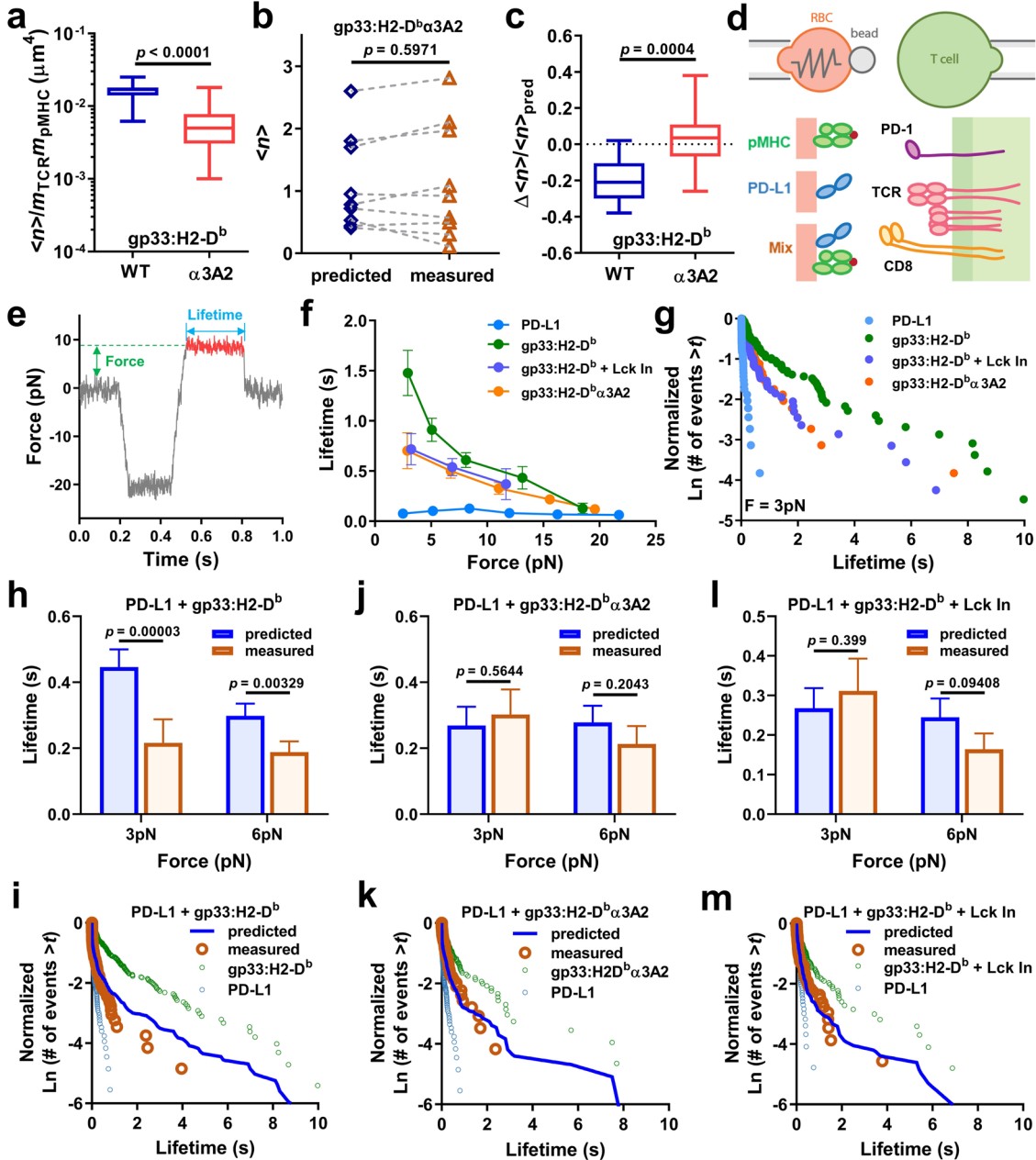

**Fig. 4 Negative cooperativity can be explained by the PD-1 suppression of TCR-CD8 cooperativity. a** Comparison of normalized numbers of bonds formed between P14 CD8+ T cells and RBCs bearing gp33:H2-D$^b$ ($n = 15$ cells) or gp33:H2-D$^b$α3A2 ($n = 21$ cells) at 2-s contact time. **b** Representative experiment comparing predicted and measured <*n*>'s determined from binding of T cells to RBCs bearing gp33:H2-D$^b$α3A2, PD-L1, or both. **c** Comparison of normalized bond reductions (Δ<*n*>/<*n*>$_{pred}$) between the gp33:H2-D$^b$ ($n = 17$ cells) and gp33:H2-D$^b$α3A2 ($n = 12$ cells) groups. **d** Schematics of the BFP setup (top) and the molecules to be analyzed (bottom). As an advanced version of the micropipette system, the BFP uses a glass bead to present pMHC, PD-L1, or both (left) to interact with the TCR, CD8, PD-1, or all of them (right). **e** Representative force-time trace recorded in a test cycle of a force-clamp assay. The level of clamped force and the duration of bond lifetime are indicated. **f** Plots of mean ± s.e.m. lifetime vs. mean force of single bonds of PD-L1, gp33:H2-D$^b$, or gp33:H2-D$^b$α3A2 interaction with untreated P14 CD8+ T cells and of gp33:H2-D$^b$ interaction with Lck inhibitor-treated P14 CD8+ T cells. **g** Bond lifetime distributions of the molecular interactions in **f** at 3 pN showing as survival probability in semi-log plots. **h–m** Comparisons between predicted and measured mean ± s.e.m. bond lifetimes (**h**, **j**, **l**; 3- and 6-pN bins) and their distributions (**i**, **k**, **m**; 3-pN bin). Untreated (**h–k**) or Lck inhibitor-treated (**l**, **m**) CD8+ T cells expressing P14 TCR and PD-1 were analyzed by the force-clamp assay against BFP beads bearing PD-L1 plus gp33:H2-D$^b$ (**h**, **i**, **l**, **m**) or PD-L1 plus gp33:H2-D$^b$α3A2 (**j**, **k**) at 2-s contact time. The predictions were based on the percentages of single-bond events formed by each of the two ligands in each lifetime ensemble, as indicated for each pair of panels (**h–j** and **k–m**), which were determined from the ligand densities coated on each group of beads and the TCR and PD-1 expressed on each batch of T cells. The bond lifetime distributions of two single-ligand bonds were also shown in **i**, **k** and **m** for comparison. The weighted sum of these single-specie bond lifetimes (distributions) were used to calculate the predicted lifetime (distribution). For the box-whisker plots in **a** and **c**, the center line labels median, the box contains the two middle quantiles, and the whiskers mark the min and the max. *p* values were calculated using Mann–Whitney test (**a**, **c**), paired *t*-test (**b**), or standard *t*-test (**h**, **j**, **l**). The sample sizes of bond lifetime events are summarized in Supplementary Table 1.

assay was performed in repetitive cycles sampling single-bond lifetimes defined as the duration of the clamped phase of a cycle where force is constant (Fig. 4e).

Approximately 2000 bond lifetimes were pooled and binned according to the force levels to generate four curves of force vs. lifetime of PD-L1, gp33:H2-D$^b$ and gp33:H2-D$^b$α3A2 with untreated T cells and of gp33:H2-D$^b$ with T cells treated with the Lck inhibitor (Fig. 4f). The average bond lifetime measured by the gp33:H2-D$^b$ probes displays a monotonic decay from 1.5 to 0.2 s as force increased from 3 to 18 pN, a pattern known to reflect the formation of "slip-bonds". Focusing on the lifetime events in the first force bin (~3 pN), histogram analysis reveals the presence of long lifetimes above average with >10% of bonds having >1 s lifetimes (Fig. 4g). We note that the lifetime pool for gp33:H2-D$^b$ probes consists of three types of molecular interactions: TCR–pMHC and CD8–pMHC bimolecular bonds and TCR–pMHC–CD8 trimolecular bonds. Due to the low affinity and short bond lifetime of the CD8–pMHC interaction, the TCR–pMHC interaction dominates the bimolecular bonds and drives the formation of the more stable TCR–pMHC–CD8 trimolecular bonds in a signaling-dependent manner[24,26]. Consistently, eliminating TCR–pMHC–CD8 trimolecular interactions by inhibiting Lck significantly reduced the average bond lifetime (0.7 vs. 1.5 s at 3 pN; Fig. 4f). Comprising of only TCR–pMHC and CD8–pMHC bimolecular bonds, the lifetime histogram also shifted toward left with fewer long-lived bonds (Fig. 4g). Similarly, when using gp33:H2-D$^b$α3A2 probes to eliminate CD8 binding, the pool contains only TCR–pMHC bonds with lifetimes much shorter than those generated using gp33:H2-D$^b$ probes (Fig. 4f, g). Moreover, the force vs lifetime curve and the lifetime histogram of Lck inhibitor-treated T cells probed by gp33:H2-D$^b$ are indistinguishable from their respective counterparts of untreated T cells probed by gp33:H2-D$^b$α3A2 (Fig. 4f, g), further confirming the need of Lck activity for TCR–pMHC–CD8 trimolecular interactions and the minimal contribution from the CD8–pMHC bimolecular interaction. In sharp contrast to the gp33:H2-D$^b$α3A2–TCR bonds, the PD-1–PD-L1 bonds have much shorter lifetimes across the force range tested, with the largest differences seen at low forces (0.08 vs. 0.70 s at 3 pN, Fig. 4f, g). The large difference in bond lifetime would allow us to resolve the altered occurrence of short vs long bond lifetime events when both pMHC and PD-L1 are co-presented to the T cells.

Similar to the bond number cooperativity analysis, the predicted bond lifetime (distribution) for the dual-ligand probes is calculated as the weight sum of bond lifetimes (distributions) from single-ligand probes assuming that prior formation of one bond species does not affect the formation of the other bond species. When the two ligands were mixed with predicted fractions of forming 29% and 71% of gp33:H2-D$^b$ and PD-L1 bonds, respectively, the predicted average bond lifetimes were 0.45 s at 3 pN and 0.30 s at 6 pN. Yet, the measured dual-species bond lifetimes were significantly shorter: 0.22 s at 3 pN and 0.19 s at 6 pN (Fig. 4h), again, showing similar negative cooperativity as in the previous bond number analysis. The bond lifetime histogram of dual-ligand probes was also left-shifted away from the predicted distribution, revealing the suppression of longer-lived (i.e. TCR–pMHC and/or TCR–pMHC–CD8) bonds (Fig. 4i). To further elucidate the role of CD8, we tested probes co-presenting gp33:H2-D$^b$α3A2 and PD-L1. No significant difference between predicted and measured average bond lifetimes or their distributions were observed (Fig. 4j, k), implying that the reduced fraction of long-lived lifetimes from the gp33:H2-D$^b$ probes comes most likely from the TCR–pMHC–CD8 trimolecular bonds. This is further confirmed by analyzing Lck inhibitor-treated T cells with probes co-presenting gp33:H2-D$^b$ and PD-L1.

Consistent with its effect of abolishing TCR–pMHC–CD8 trimolecular bond formation and the negative cooperativity in bond number analysis, the negative cooperativity in terms of bond lifetime also diminished (Fig. 4l, m). These data further reveal that PD-1 suppresses the signaling-dependent cooperative binding of TCR and CD8 to pMHC manifesting negative cooperativity in their concurrent binding to dual-ligands.

**PD-1's suppression of TCR-CD8 cooperativity requires CD8–Lck association.** Our proposed molecular mechanism of positive cooperative binding of pMHC by TCR and CD8 ecto-domains involves a TCR proximal signaling-dependent "Lck bridge" that on one end associates with CD8 via its N-terminus and, on the other, docks on phosphorylated CD3 ITAMs[24–26], similar to that in TCR-CD4 coupling[29,30,45,46]. As PD-1 inhibits the early activating signals including the phosphorylation of CD3ζ and ZAP70[6,8,12], we hypothesize that the negative cooperativity is due to the disruption of this Lck-mediated dynamic molecular assembly. To test this, we sorted CD8⁻CD4⁻ (double negative, DN) thymocytes (Fig. 5a) from P14 mice and transduced the cells with retrovirus encoding CD8αβ WT or C227SC229S mutant, where the Lck-binding C$^{227}$KC$^{229}$P motif of the α chain is replaced with S$^{227}$KS$^{229}$P to eliminate Lck association[25,47] (Fig. 5b). Consistent with observations in other TCR-transgenic mice[48,49], CD44 staining reveals most of the P14 DN thymocytes are in DN3 and DN4 stages (CD44⁻, Supplementary Fig. 3a), where Lck is expressed[50]. The expression of CD8, TCR, and PD-1 are similar on CD8WT and CD8SKSP cells (Fig. 5b and Supplementary Fig. 3b, c). When tested against the same RBCs bearing gp33:H2-D$^b$, DN thymocytes re-expressing CD8WT formed more bonds than those re-expressing CD8SKSP that abolished cytoplasmic Lck association (Fig. 5c), consistent with our previous results suggesting that the cooperative TCR–pMHC–CD8 trimolecular interaction requires CD8–Lck association[25]. When tested against the same DN thymocytes re-expressing CD8WT, RBCs bearing gp33:H2-D$^b$ formed significantly higher number of bonds than those bearing gp33:H2-D$^b$α3A2, confirming the synergistic CD8 binding (Fig. 5d). However, when tested against the same DN thymocytes not expressing CD8 or re-expressing CD8SKSP, the differences in the number of bonds formed with RBCs bearing gp33:H2-D$^b$ and gp33:H2-D$^b$α3A2 were no longer observed, confirming the contribution of CD8 synergy (Fig. 5d). Most importantly, analysis of dual-ligand (gp33:H2-D$^b$ and PD-L1) binding shows a bond number reduction in CD8WT but not CD8SKSP cells (Fig. 5e). These data further indicate the adapter role of Lck in the "inside-out" regulation of TCR-CD8 positive cooperation and its suppression by PD-1.

**PD-1 suppresses TCR-CD8-triggered Ca$^{2+}$ signaling.** To evaluate the functional consequence of the PD-1 suppression of antigen recognition by the TCR–pMHC–CD8 interaction, we analyzed the Ca$^{2+}$ response to pMHC stimulation in the presence or absence of PD-L1. *PDCD1*$^{-/-}$ P14 T cells re-expressing PD-1 generated robust Ca$^{2+}$ flux upon landing on surface coated with gp33:H2-D$^b$ (Fig. 6a, b). About 90% of cells have an amplitude above 1.5-fold of baseline with earliest Ca$^{2+}$ flux seen within 1 to 2 min upon landing (Fig. 6a, b, e). In contrast, additional PD-L1 coating on the surface significantly reduced the Ca$^{2+}$ signal (Fig. 6c–e). Comparing to the condition with gp33:H2-D$^b$ only, amplitude contours of the pseudo-image (Fig. 6d) and distribution analysis reveal much smaller percentage of cells reaching the same level of Ca$^{2+}$ flux when PD-L1 was present (Fig. 6e). The maximum and time-integrated signals were also significantly lower (Fig. 6f, g). These data indicate reduced downstream

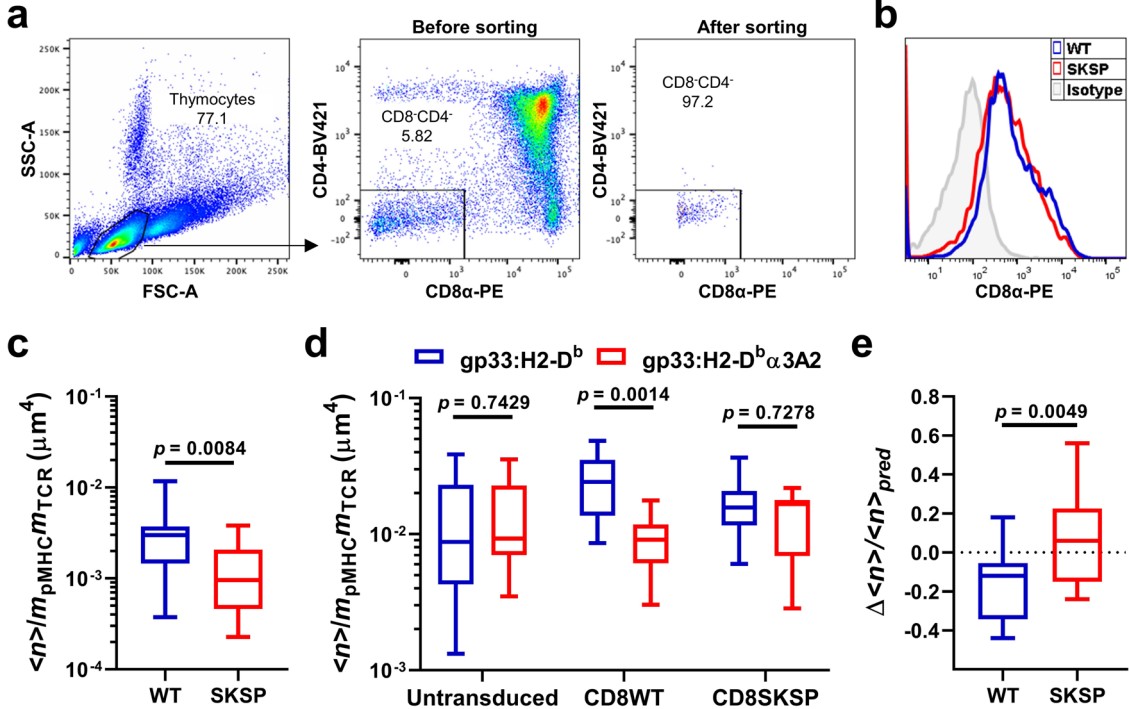

**Fig. 5 PD-1's suppression of TCR-CD8 cooperativity requires CD8–Lck association.** Sorted P14 CD8⁻CD4⁻ (DN) thymocytes were transduced to express CD8 WT or SKSP mutant that abolishes Lck binding, followed by 2D kinetic analysis for negative cooperativity. **a** Representative CD4 vs CD8 plots of total P14 thymocytes showing the sorting strategy (left and middle) and purity (right) of DN thymocytes. **b** Representative histogram plot of retroviral-transduced P14 DN thymocytes expressing CD8 WT or SKSP mutant. **c** Comparison of normalized numbers of bonds formed between RBCs bearing gp33:H2-D$^b$ and P14 DN thymocytes transduced to express CD8WT or CD8SKSP ($n = 17$ and 15 cells) at 2-s contact time. **d** Comparison of normalized numbers of bonds formed between untransduced, CD8WT, or CD8SKSP P14 DN thymocytes and RBCs bearing gp33:H2-D$^b$ ($n = 11$, 11, and 11 cells) or gp33:H2-D$^b$α3A2 ($n = 10$, 10, and 12 cells) at 5-s contact time. **e** Comparison of normalized bond reductions (Δ$<n>$/$<n>$$_\text{pred}$) between CD8WT and groups CD8SKSP ($n = 16$ and 17 cells). Data are presented by box-whisker plots (the center line labels median, the box contains the two middle quantiles, and the whiskers mark the min and the max) with the $n$ values indicating the numbers of cells measured per condition. $p$ values were calculated using Mann–Whitney test.

signaling when the initial antigen recognition was perturbed by PD-1.

## Discussion

Since the discovery of PD-1 in 1992[51], decades of effort have lead us to an understanding of how PD-1 inhibits TCR and CD28 signaling, thereby controlling activation and regulating the function and metabolism of conventional T cells[52]. Its critical role in T cell exhaustion also has brought us the great opportunity of targeting PD-1 or its ligands to rescue lost T cell functions for immunotherapies of cancer or potentially chronic infectious diseases[1–3,53]. Despite the rapidly expanding clinical applications, many questions remain unclear regarding the mechanism of PD-1 triggering and signaling. In this study, we investigated the effect of PD-1 on T cell antigen recognition – the initiation of all T cell activation and subsequent responses. By combining imaging and in situ kinetic analysis, we observed a negative cooperativity between PD-1 and the TCR-CD8 axis in their concurrent ligand binding. This manifests as reduced cell spreading on, less bond formation and shorter bond lifetime with pMHC, indicating the disruption of the molecular interactions for antigen recognition.

In conventional binding cooperativity, as that typically takes place with isolated molecules, binding of a second molecule is often physically modulated by binding of the first molecule. By comparison, in situ cooperative binding of cell surface molecules may occur by direct physical interplay of their ectodomains, and/or by inside-out regulation through signaling events that involve additional intracellular and/or membrane molecules. An example

for the former case is the recently discovered *cis* heterodimerization between PD-L1 and B7-1 on APC surface, which prevents trans-interaction with PD-1 but not CD28, the receptor for B7-1[54,55]. This negative cooperativity is due to the masking of PD-1 binding site on PD-L1, but not the CD28 binding site on B7-1, upon the PD-L1–B7-1 *cis*-interaction. Our results provide an example for the latter case, as the elimination of the negative cooperativity by inhibiting SHPs or Lck cannot be explained by ectodomain *cis*-interactions between PD-L1 and H2-D$^b$ or between PD-1 and TCR or CD8, none of which have been reported. Higher-level physical interference due to mismatch dimensions of the two *trans*-interactions is also unlikely, because the negative cooperativity was abolished by the H2-D$^b$α3A2 mutant that spans the same dimension as H2-D$^b$. Instead, the negative cooperativity indicates an "inside-out" signaling feedback, where the interplay of the intracellular signals triggered by concurrent ligand binding of PD-1 and TCR-CD8 negatively regulates the extracellular cooperation of TCR and CD8 in binding to pMHC. A well-studied example of this type of loop is the "inside-out" upregulation of integrin ligand-binding, where activating signals from other receptors induce conformational changes and/or cell surface clustering of integrin molecules from low- to intermediate- and high-affinity/avidity states to mediate enhanced cell adhesion on the ligand presenting surface[34,56–58]. In addition to changes in molecular structures, more complex in situ cooperative binding could stem from global cell behaviors that integrate functions of multiple receptors. Pielak et al. reported an upregulation of CD28–B7-1 interaction by TCR–pMHC ligation during T-cell formation of IS with ligand-

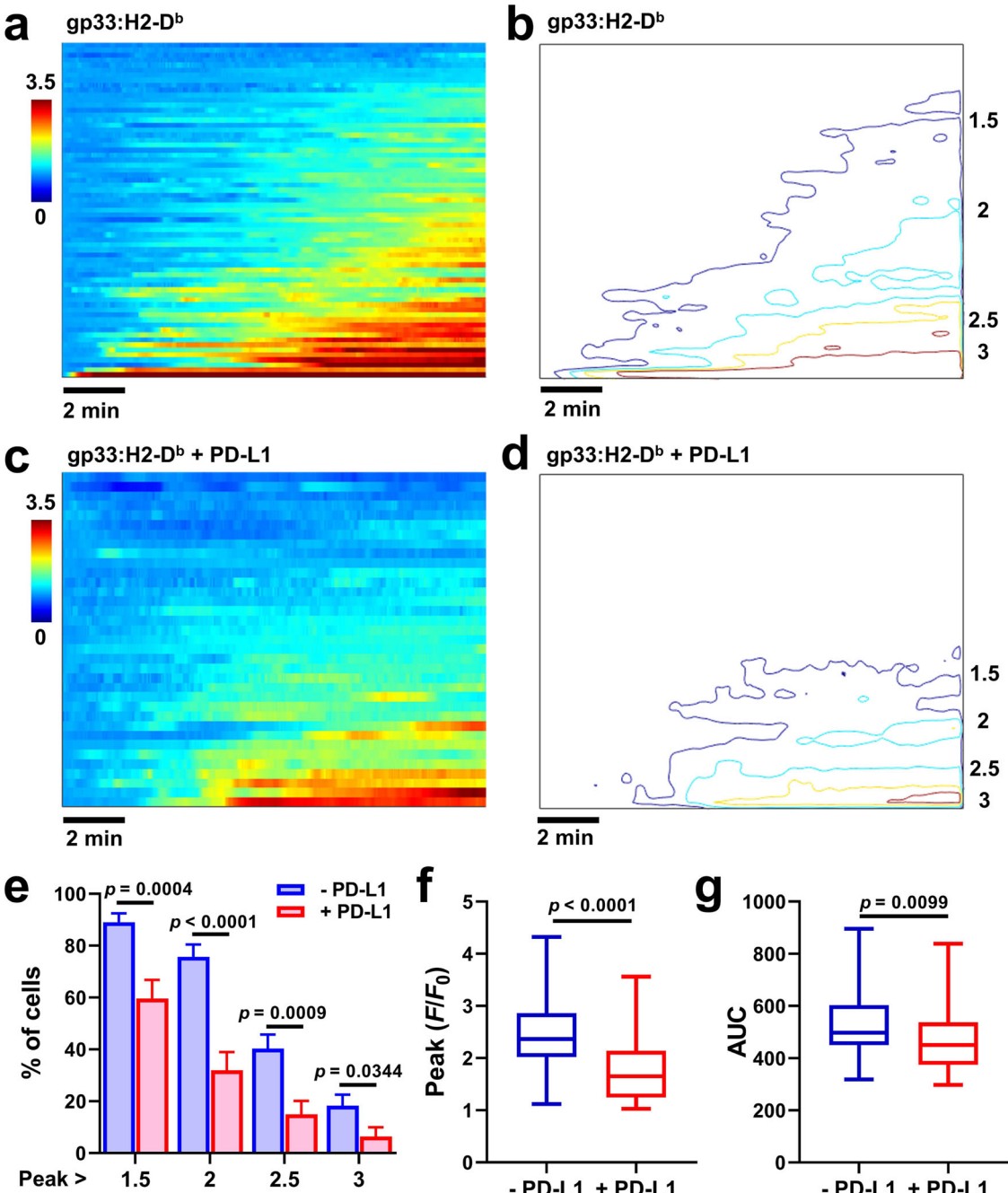

**Fig. 6 PD-1 suppressed antigen-triggered T cell Ca$^{2+}$ signaling. a, c** Heat maps constructed using Ca$^{2+}$ images from individual cells to show the signal dynamics. Activated *PDCD1$^{-/-}$* P14 CD8$^+$ T cells re-expressing PD-1 were loaded with the X-Rhod-1 calcium dye, placed on glass coverslips coated with gp33:H2-D$^b$ alone (**a**, $n = 82$ cells) or gp33:H2-D$^b$ plus PD-L1 (**c**, $n = 47$ cells), and imaged under a microscope with a 2-s frame interval for 20–30 min. Each row represents an individual cell aligned by their landing time to the glass surface and sorted from top to bottom with increasing Ca$^{2+}$ signal strength. The pseudo-color represents normalized Ca$^{2+}$ signal against base fluorescence. **b, d** Contour plots of images in **a** and **c** at indicated fold-increase. **e–g** Quantification of **a–d** showing percentages of cells (mean ± s.e.m.) with calcium peak above indicated levels of fold-increase (**e**), and box-whisker plots of the peak values (**f**) and area under curves (AUC) (**g**) of calcium time courses of individual cells with the center line labeling median, the box containing the two middle quantiles and the whiskers marking the min and the max. *p* values were calculated using standard *t*-test (**e**) or Mann–Whitney test (**f, g**).

functionalized lipid bilayer[59]. This positive cooperativity seems to be a global effect on CD28 at the intercellular interface due to TCR-triggered integrin activity, possibly by creating a tighter cell–cell junction and therefore enhancing the on-rate of 2D binding[59].

Our data showing PD-1 suppression of T-cell spreading suggests the possibility of a similar global regulation of intercellular junction during antigen recognition, as it may involve TCR-signaling-

dependent cytoskeleton activity in the spreading process. Studies of T-cell motility in vivo or interactions with functionalized lipid bilayer or with APC in vitro also demonstrated the PD-1 negative regulation at a global level. Blockade of PD-1 or PD-L1 reduced T cell motility and promoted antigen-induced T cell arrest in an autoimmune diabetic model and a skin inflammation model, which was due to relieving PD-1 inhibition of the TCR-triggered "stop signal"[60,61]. PD-1 disruption of stable synapse formation

in vitro was also reported for CD8[+] T cells from chronic lymphocytic leukemia models[62,63] or CD4[+] T cells from AND TCR transgenic mice[8]. The impairment was possibly due to PD-1's suppression of a TCR-triggered upregulation of integrin binding to their ligands[64]. Yet, this kind of PD-1 regulation could be context-dependent, as PD-1 was also found to stabilize IS formation of P14 T cells from LCMV infection and restrain T cell motility in virus-infected spleen[12].

It is noteworthy that, unlike the IS studies where the interactions between two cells or between a cell and a surface were largely mediated by adhesion molecules like integrins, there was no integrin binding involved in our dual-ligand presentation of pMHC and PD-L1. Cell spreading was due only to the interactions of TCR-CD8 and PD-1 with pMHC and PD-L1, respectively, making it possible to attribute the observed PD-1 effects at least partly to the downregulation of these molecular interactions. This was then confirmed in our 2D kinetic analysis where T cell contact with a surrogate APC was externally controlled instead of resulting from the cells' own action. The 2D kinetic analysis also underlined the molecular structures or their surface organizations that determine the sensitivity and responsiveness of a receptor to a specific "inside-out" signaling regulation. The lack of effect for SHPs or Lck inhibitors on PD-1–PD-L1 binding might be partially explained by PD-1's relatively simple structure, which consists of a single Ig V-like domain connecting to the ITIM and ITSM-containing cytoplasmic tail functioning as a monomer[10,65]. In contrast, both the number and the lifetime of the TCR–pMHC–CD8 bonds were greatly reduced when Lck kinase activity was abolished, recapitulating the previously reported Lck-dependent TCR-CD8 cooperative binding of pMHC[17,24–26]. This positive cooperativity of extracellular binding relies on CD8–Lck association and the phosphorylation of CD3 ITAMs and possibly other proximal signaling molecules, suggesting a dynamic molecular assembly that may share similar structure and composition for the TCR-CD4 coupling via CD4-associated Lck docking on phosphorylated CD3 ITAMs or the ITAM-bound ZAP70[27–31,45,46]. These distinct properties of PD-1 and TCR-CD8 therefore suggest that each interaction could be regulated differently when crosstalk of the two signaling axes occurs.

Using a mutant MHC, we were able to eliminate the negative cooperativity by abolishing binding of the CD8 ectodomain, which corroborates the evidence obtained by disrupting CD8–Lck association and by perturbing the intracellular signaling apparatus. Applying force spectroscopic analysis, we resolved the lifetime differences among PD-1–PD-L1 (short), TCR–gp33:H2-D[b] (intermediate), and TCR–gp33:H2-D[b]–CD8 (long) bonds. This analysis has provided more evidence in addition to that from the bond number analysis: the skewed occurrence of more short-lived and fewer long-lived bonds when we tested T cells using probes co-presenting gp33:H2-D[b] but not gp33:H2-D[b]α3A2 with PD-L1. Finally, both TCR-CD8 positive cooperativity and TCR-PD-1 negative cooperativity were eliminated when the intracellular association between CD8 and Lck were disrupted by mutations. Together, our data suggest that the PD-1 suppression of TCR-CD8 positive cooperativity for pMHC binding as the mechanism for the negative cooperativity.

This work adds PD-1 and SHPs to the TCR interaction network model[22] built on the data that CD3 phosphorylation and Lck are required for the TCR-CD8 cooperative binding of pMHC[24–26]. Studies disrupting Lck association with CD8/CD4 or inhibiting its activity suggested that the initial step of TCR triggering is likely mediated by Lck not associated with co-receptors[25,31,66] (Supplementary Fig. 4a). Instead, phosphorylated CD3 (and ZAP70) may serve as docking sites for CD8/CD4-associated Lck, recruiting CD8/CD4 to the TCR to augment pMHC binding[29,30,67,68] (Supplementary Fig. 4b). The binding

cooperativity reinforces TCR signaling by prolonging pMHC engagement, sustaining Lck co-localization, and possibly regulating Lck's activity. The assembly could be highly dynamic due to the dependency on the accumulation of proximal signaling, and possibly the exchange of Lck among co-receptor-associated, membrane-bound, CD3-associated[69], and free states. The reliance of this mechanism on TCR proximal signaling also renders the augmentation by positive feedback sensitive to PD-1's inhibitory effect (Supplementary Fig. 4c). Although PD-1-associated SHP2 preferentially dephosphorylates CD28 in a cell-free system and in Jurkat cells[13], phosphatases like SHP1 and SHP2 generally do not show high substrate specificity[71]. PD-1 was shown to significantly inhibit the phosphorylation of CD3 and ZAP70 in other T cell lines or primary T cells[6,8,12] and reduce cytokine production in the absence of a CD28 signal[70]. Lck and PD-1 were shown to cross-regulate the phosphorylation of each other in a cell-free system[13], but how Lck activity could be affected by PD-1 in cells remains unclear and may depend on the location of Lck and PD-1 as well as other Lck regulators[72]. We speculate that PD-1 may also inhibit the accumulation of ZAP70 on phosphorylated CD3 and the recruitment of Lck, thereby disrupting the molecular bridge for CD8 recruitment (Supplementary Fig. 4c). In this way, PD-1 signaling can feedback on the most upstream of the TCR signal initiation process, inhibiting T cell functions at a very early stage. These observations help understand the potent inhibitory effect of PD-1 on T cell functions and provide a basis for future optimizing PD-1-based therapeutic interventions.

## Methods

**Mice and cells**. In all, 6–8-weeks-old female C57BL/6 mice were purchased from The Jackson Laboratory. P14 transgenic, PDCD1[−/−] P14 transgenic, and C57BL/6 mice were housed at the Emory University Department of Animal Resources facility (Daylight cycle 7 a.m. to 9 p.m.; Darklight cycle 9 p.m. to 7 a.m.; Ambient Temperature 72 °F; Humidity 30–70%.) and used in accordance with National Institutes of Health and the Emory University Institutional Animal Care and Use Committee guidelines. For experiments using splenic CD8[+] T cells, total splenocytes were prepared by mechanical grinding of the spleen followed by RBC lysis (eBiosciences) according to the manufacturer's instructions. P14 splenocytes were incubated at a density of $2 \times 10^6$ cells/ml for 2 h at 37 °C with 10 nM LCMV gp$_{33–41}$ (KAVYNFATM). Cells were then washed with HBSS, resuspended in R10 medium (RPMI 1640 supplemented with 10% FBS, 100 U/mL penicillin, 100 μg/mL streptomycin, 2 mM L-glutamine, 20 mM HEPES, and 50 mM 2-Mercaptoethanol) and cultured at $4 \times 10^6$ cells/3 ml/well in a 12-well plate at 37 °C with 5% CO$_2$. CD8[+] T cells were purified on day 2 or 3 post activation via Ficoll gradient separation followed by CD8 negative purification using EasySep™ Mouse CD8[+] T Cell Isolation Kit (Stemcell Technology). For experiments using DN thymocytes, total thymic cells were prepared by mechanical grinding and stained with PE-conjugated anti-CD8a (1:100 dilution) and BV421-conjugated anti-CD4 (1:100 dilution) antibodies. DN thymocytes were sorted by gating on CD8[−]CD4[−] population with a purity of ~97% (Fig. 5a) and cultured in R10 medium supplemented with 1 ng/ml recombinant mouse IL-7. Human RBCs were isolated from healthy donors and used according to a protocol approved by the Institutional Review Board of the Georgia Institute of Technology.

**LCMV infection**. To generate P14 chimeric mice, we transferred $1 \times 10^3$ P14 cells from spleen of P14 mice (Thy1.1/1.1) into naïve B6 (Thy1.2/1.2) mice. The P14 chimeric mice were infected with LCMV Armstrong (LCMV Arm) ($2 \times 10^5$ p.f.u. i.p.) for acute viral infection or infected with LCMV clone13 (LCMV CL13; $2 \times 10^6$ p.f.u. i.v.) for chronic viral infection. Mice were sacrificed at indicated days and P14 cells were sorted based on Thy1.1 staining.

**Retroviral transduction of splenic T cells and thymocytes**. We reconstituted PD-1 expression in activated PD-1 KO P14 cells and CD8WT or CD8SKSP mutant in DN thymocytes via retroviral transduction as previously described[73–75]. Briefly, mouse PD-1 or CD8α and CD8β chains joint by a P2A element was cloned into pMSCV-IRES-GFP II vector (pMIG II, Addgene # 52107). CD8SKSP mutant was generated by mutating both C227 and C229 of CD8WT α chain into Serine residues. To produce retrovirus, 293T cells (ATCC) were transfected with packaging plasmid pCL-Eco (Addgene # 12371) together with pMIG II or pMIG II containing mPD-1, CD8WT, or CD8SKSP using Lipofectamine 3000 (ThermoFisher Scientific) following manufacture's protocol. Cell culture medium (DMEM supplemented with 10% FBS, 6 mM L-glutamine, 0.1 mM MEM non-essential amino acids, and 1 mM sodium pyruvate) was replaced after overnight culture.

Supernatant containing retrovirus was harvested 48–72 h later and froze at −80 °C until use. For transducing splenic T cells, naïve CD8$^+$ T cells purified from RBC-lysed splenocytes of PD-1 KO P14 mice using EasySep™ Mouse CD8$^+$ T Cell Isolation Kit were activated with plate-bound anti-CD3 (clone 145-2C11, Tonbo Biosciences), 0.5 µg/ml anti-CD28 (clone 37.51, Tonbo Biosciences), and 100 U/ml recombinant human IL-2 (Gemini Bio) for 24 h. Activated T cells were Percoll (GE Healthcare) purified and spinoculated for 1.5 h in 1:1 mix of R10 and retroviral supernatant containing 0.5 µg/ml anti-CD28, 100 U/ml recombinant human IL-2 and 8 µg/ml polybrene (Sigma-Aldrich) at $3 \times 10^6$ cells/ml. Cell culture medium was replaced the next day with complete R10 with 100 U/ml recombinant human IL-2. GFP$^+$ cells were sorted using BD Fusion cell sorter (BD Biosciences) day 3 or 4 after transduction. Sorted cells were allowed to rest for at least 24 h before use. For transducing DN thymocytes, retroviral supernatant was added to retronectin-coated plate and centrifuged for 1.5 h. Due to the decrease of endogenous PD-1 expression during in vitro culture, PD-1 retrovirus was mixed with CD8WT or CD8SKSP retrovirus for experiments of analyzing PD-1's effect. Sorted CD8$^−$CD4$^−$ thymocytes were rested in R10 medium containing 1 ng/ml recombinant IL-7 for 3 h before being added to virus-coated plate. Cells were cultured overnight before flow cytometry and 2D kinetics analyses. A second round of transduction was performed as needed to boost the expression of CD8 and PD-1.

**Proteins and chemicals.** His6-tagged mPD-L1 with BirA sequence were produced in CHO cells (ATCC) as described previously[76]. Biotinylation was performed in vitro using the BirA biotin-protein ligase kit (Avidity). Recombinant mouse ICAM-1 with a human IgG1 Fc tag was from Biolegend. Wide type and α3A2 mutant of gp33/H2-D$^b$ were made by the National Institutes of Health Tetramer Core Facility at Emory University.

PE-conjugated anti-mPD-1 (clone J43, 1:20), anti-mTCR Vα2 (clone B20.1, 1:20), anti-mCD8α (clone 53-6.7, 1:20), isotype RatIgG2a,λ (clone B39-4, 1:20), isotype RatIgG2a,κ (clone A95-1, 1:20), and isotype american hamster IgG2,κ (clone B81-3, 1:20) were from BD Biosciences. PE-conjugated anti-mPD-L1 (clone MIH5, 1:20) and biotinylated anti-human IgG1 (clone HP6070, 1:200) were from ThermoFisher Scientific. BV421-conjugated anti-CD4 (clone GK1.5, 1:100), PE-conjugated anti-mCD8α (clone 53-6.7, 1:100), anti-H2-D$^b$ (clone KH95, 1:20), and isotype ratIgG1,κ (clone RTK2071, 1:20) were from Biolegend. APC-conjugated anti-CD44 (clone IM7, 1:20) was from Tonbo biosciences. Biotinylated anti-His tag (1:40) was from Qiagen.

SHP1 and SHP2 inhibitor NSC87877 was from Santa Cruz. Lck inhibitor 7-Cyclopentyl-5-(4-phenoxyphenyl)-7H-pyrrolo[2,3-d]pyrimidin-4-ylamine was from Sigma Aldrich.

**Flow cytometry.** Samples were stained in 100 µl of FACS buffer (PBS without Ca$^{2+}$ or Mg$^{2+}$ supplemented with 5 mM EDTA and 2% FBS) containing 10 µg/ml (or as indicated above) of antibodies for 30 min at 4 °C. Sample were washed twice with 2 ml of FACS buffer, fixed with 200 µl of 1% PFA for 15 min at 4 °C. Fixed samples were washed once with 2 ml of FACS buffer, and resuspended in 300 µl of FACS buffer for analysis under LSR II flow cytometer (BD Biosciences). Flow cytometric data were analyzed using FACS DIVA (BD Biosciences) and FlowJo (TreeStar).

**Cell spreading assay.** In all, 96-well imaging plate was cleaned with 1 N NaOH followed by thorough wash with diH$_2$O. The imaging surface was then prepared in the following incubation steps with three washes using PBS after each step: (1) coating with 50 µg/ml biotinylated bovine serum albumin (BSA); (2) coating with streptavidin; (3) coating with mixture of biotinylated pMHC and biotinylated anti-PentaHis (Qiagen) or biotinylated anti-human IgG1 Fc; and (4) coating his-tagged mPD-L1 or human IgG1-tagged mICAM-1. T cells from various groups were resuspended in imaging buffer (HBSS with Ca$^{2+}$ and Mg$^{2+}$ supplemented with 2% FBS), added to the surface and incubated for 20 min at room temperature. Cells were then imaged under Nikon Ti microscope equipped with a ×60 TIRF objective, a RICM cube, and a mercury lamp for excitation at 560 nm. Images were acquired using Volocity (PerkinElmer) with cell spreading area calculated using ImageJ and customized Matlab (Mathworks) scripts by thresholding the RICM images for dark regions.

**Ca$^{2+}$ imaging.** Imaging surfaces were prepared as described above for cell spreading assay. T cells at a density of $1 \times 10^6$ cells/ml were incubated with 5 µM X-Rhod-1 (ThermoFisher Scientific) in R10 medium for 30 min at 37 °C. Cells were washed twice with imaging buffer and immediately added onto ligand-coated surface. Upon addition, cells were imaged under an Olympus IX70 microscope equipped with a ×20 air objective. Cells were excited with a Xenon lamp at 580/15 nm and emission was acquired at 620/60 nm with a 2-s interval for 20–30 min. The image stack was collected using Micro-Manager[77] and analyzed using customized Matlab application to calculate the fluorescence changes over time for each cell. Briefly, cells on each frame were defined upon adaptive thresholding and tracked for the entire stack to calculate the fluorescence traces for each cell. After determining the cell landing frame ($t_0$), normalized fluorescence against baseline was calculated to reflect the fold change. Normalized fluorescence traces were aligned by $t_0$ and sorted based on their maximum fold change to generate the pseudo-image as shown in Fig. 6a, c.

**Micropipette adhesion frequency assay.** The theoretical framework and detailed procedures of this assay have been reported previously[16,35,36]. Briefly, a RBC was used as a surrogate APC to present ligands and as a force transducer to detect binding to a T cell. RBCs were isolated from whole blood drawn from healthy donors according to a protocol approved by the Institute Review Board of Georgia Institute of Technology. RBCs were biotinylated using various concentrations of biotin-NHS linker (ThermoFisher Scientific). Biotinylated RBCs were first coated with saturating amount of streptavidin (SA) and washed. SA-coated RBCs were then incubated with saturating amount of biotinylated recombinant proteins and washed before the experiments. During the experiment, a T cell was aspirated by a piezo-driven micropipette whose movement was precisely controlled using Labview (National Instruments) programs. Each T cell was repeatedly brought into contact with a ligand-coated RBC and held for a pre-defined duration ($t_c$). Adhesion was detected during the separation of the two cells from the membrane stretch of the RBC. The adhesion frequency ($P_a$) was determined after 30–50 cycles and was used to calculate the average number of bonds per contact $<n>$ and the effective 2D affinity as follows:

$$P_a = 1 - \exp(-<n>) \tag{1}$$

$$\text{and} \quad <n> = m_r m_l A_c K_a [1 - \exp(-k_{off} t_c)]. \tag{2}$$

Here $m_r$ and $m_l$ are the respective densities of the receptor on the T cell and the ligand on the RBC that were measured using PE-labeled monoclonal antibody together with QuantiBRITE PE standard beads (BD Biosciences), $A_c$ is the contact area, $K_a$ is the 2D affinity (in µm$^2$), and $k_{off}$ is the off-rate (in s$^{-1}$). With long contact duration (e.g. 5 s), $P_a$ and $<n>$ approach equilibrium, and the effective 2D affinity $A_c K_a$ was estimated by normalizing $<n>$ against receptor and ligand densities.

$$A_c K_a = <n> / m_r m_l \tag{3}$$

**Biomembrane force probe force-clamp assay.** The procedures for BFP force-clamp assay of single-bond lifetime under force have been described previously[19]. Briefly, a T cell was aspirated by a piezo-driven micropipette whose movement was precisely controlled using Labview (National Instruments) programs. Each t cell was repetitively brought into contact with a ligand-coated glass bead attached to the apex of a micropipette-aspirated RBC, which serves as an ultrasensitive force transducer. After contact, the T cell was retracted and held at a distance corresponding to the set force level. The displacement of the bead tracked at 1000 fps was translated into force reading with a preset RBC spring constant of 0.3 pN/nm. Molecular bond formed between the ligand on the bead and the receptor on the T cell pulls the bead away from its original position during T cell retraction, as reflected by the increase in force applied on the bond. The force (bead displacement) sustains until the bond ruptures, with the total duration defining the bond lifetime under the clamped force level. Repeated measurement cycles at multiple force levels generated a pool of such events, which were presented in the form of average bond lifetime $<t>$ vs average force after binning. Cumulative histogram of lifetime events for each bin were also calculated as the natural log of the number of events with a lifetime $>t$.

**Cooperativity analysis of bond number and bond lifetime.** Cooperativity analysis based on comparison of the number of dual-species of receptor–ligand bonds with the sum of the numbers of the two component single-species bonds has been described previously[24,37–39]. For the molecular systems in this study, the average bond numbers for RBCs coated with individual or mixed ligands were calculated as follows:

$$<n>_{pMHC} = -\ln(1 - P_{a,pMHC}), \tag{4}$$

$$<n>_{PD-L1} = -\ln(1 - P_{a,PD-L1}), \tag{5}$$

$$<n>_{mix} = -\ln(1 - P_{a,mix}). \tag{6}$$

For concurrent and independent interactions of pMHC and PD-L1 with their respective receptors on the T cell, bond formation is governed by their affinities as defined in Eq. (3). The predicted total bond number would be the sum of the individual bond numbers[37–39]

$$<n>_{pred} = r_{pMHC} <n>_{pMHC} + r_{PD-L1} <n>_{PD-L1}, \tag{7}$$

where $r_{pMHC}$ and $r_{PD-L1}$ are the ratios of ligand densities on dual-ligand bearing RBC versus the two single-ligand bearing RBCs, all of which were measured by flow cytometry. The net cooperativity is then calculated as the difference between predicted total bond number and that measured in the mixture coating condition

$$\Delta<n> = <n>_{mix} - <n>_{pred}. \tag{8}$$

We name it apparent "positive cooperativity" or "negative cooperativity" in the case $\Delta<n> > 0$ or $\Delta<n> < 0$, respectively. The percentage of changes in bond

number is then defined as $\Delta<n>/<n>_{pred}$. To reduce cell–cell variation among groups with different coating, we tested the same T cell in randomized order against three RBCs coated with individual ligands or mixed ligands, which allowed for cooperativity analysis with paired single-cell readout.

Cooperativity analysis of bond lifetime is similar except that the readout now is the lifetimes of single bonds from all possible interactions involved. By binning the events based on their clamped force, we calculated the average bond lifetime for each coating conditions: $<t>_{pMHC}$, $<t>_{PD-L1}$, and $<t>$ mix as well as the cumulative lifetime histogram $Pt_{pMHC}$, $Pt_{PD-L1}$, and $Pt_{mix}$. The predicted average bond lifetime and cumulative histogram were defined as

$$<t>_{pred} = f_{pMHC}<t>_{pMHC} + f_{PD-L1}<t>_{PD-L1} \tag{9}$$

$$Pt_{pred} = f_{pMHC}Pt_{pMHC} + f_{PD-L1}Pt_{PD-L1}, \tag{10}$$

Where $f_{pMHC}$ and $f_{PD-L1}$ are the fractions of each species in total molecular bonds as predicted by Eq. (3).

**Statistics and reproducibility.** Replication were performed 2–3 times independently to ensure reproducibility and increase the sample size (cells or bond lifetime events). Groups with fewer than 10 cells or bond lifetime events were from single experiment. Statistical analysis was performed using Excel (Microsoft), Prism (GraphPad software), and Matlab (MathWorks). Comparison of two groups were based on two-sided Mann–Whitney test unless two-sided Student's $t$ test or paired $t$-test was noted. For Fig. 6e, bootstrapping (10,000 times) was applied to calculate the mean and standard error for each bar.

**Reporting summary.** Further information on research design is available in the Nature Research Reporting Summary linked to this article.

## Data availability

Data supporting the findings of this study are presented in the article and supplementary materials and are available from the corresponding author upon reasonable request. Source data are provided with this paper.

## Code availability

Customized codes for imaging data analysis are available from the corresponding author upon reasonable request.

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

## Acknowledgements

This work was supported by NIH grants U01CA214354 (to C.Z.), R01CA243486 (to C.Z.), and U01CA250040 (to C.Z. and R.A.).

## Author contributions

K.L., R.A., and C.Z. designed experiments; K.L., Z.Y., J.L., and E.A. performed experiments; K.L., Z.Y., J.L., E.A., and C.Z. analyzed the data. K.L., S.J.D., and C.Z. wrote the paper with contributions from other authors.

## Competing interests

The authors declare no competing interests.
