## [Peer Review File · Nature Communications]

REVIEWER COMMENTS

Reviewer #1 (Remarks to the Author):

The paper by Li and coworkers document inhibitory effects of PDL1-PD-1 ligation on CD8-TCR cooperativity. It is an interesting and potentially important study with elegant technology, but there are some issues. A central missing experiment is to analyse effects in which CD8 has been mutated to prevent the binding of Lck. There exist CD8- T-cell lines and most specifically, CD8- T-cells can be isolated from primary T-cells (also CD4-) and transduced or transfected with the mutant. At a minimum, the CD8-Lck binding mutant can be overexpressed in T-cells under conditions where it interferes with responses to gp33 presentation responses due to the absence of Lck binding, could be differentially labelled and assessed for different effects.

Specific Points

- 1) Figure 1: looks convincing but more information on numbers of cells examined is needed as well as control co-ligation with another receptor-ligand pair.
- 2) "negative cooperativity" seems an oxymoron and leads to confusion. Cooperativity cannot by definition be negative. You have cooperativity that is inhibited by PDL-1. Does the term need to be changed, perhaps "inhibited cooperativity", "induced dispersion"?
- 3) Fig 2: looks convincing. Panel g is unclear PDL1 should be replaced by no peptide PDL-1. Need control with no PDL-1 and no peptide.
- 4) How do the authors reconcile their findings with the claim by Gascoigne lab claiming that TCR signalling is triggered by CD8 not associated with Lck (Nat Commun 5, 5624 (2014).
- 5) It is not clear that the T-cells responses are CD8 dependent in terms of eliciting proliferation and the generation of cytolytic T-cell responses.
- 6) How do the authors reconcile their model with the current model arguing that PD-1 operates primarily during the effector phase of the T-cell response against antigens, in particular tumour neoantigens? Their findings may be true but of little real consequence to the real effect of PD-1 during the effector phase of the T-cells response. The authors need to address this issue. Do the mechanisms that they describe operate during the killing phase of CTLs or do they think that the other models of PD-1 modulation of responses are incorrect?
- 7) Although the SMI: 7-Cyclopentyl-5-(4-phenoxyphenyl)-7H-pyrrolo[2,3-d]pyrimidin-4-ylamine only is advertised as Lck specific, it is unclear whether it might also affect other Src kinases. One must use mutated CD8 without Lck binding (Cys mutants) in order to make their claims rigorous and specifically related to CD8 function.
- 8) Do inhibitors of ERK affect the processes in the same way as inhibitors of Lck? It would address whether the Lck requirement is specific at a proximal level or involve general calcium-dependent activation events in T-cells.
- 9) One hopes that RBC presentation events are equivalent to DC presentation events...a comment is needed.

10) Is not "binding-signaling- binding" just classic "inside out signaling"? Do we really need a new term?

11) Are not the effects of SHP2 consistent with its positive signaling role as described in most non-immunological literature?

12) Introduction: small point but still important. 1st para, line 4 should reference some of the original seminal papers on the identification of CD4 and CD8-lck from which all subsequent work originated. For example, Rudd et al (1988) Proc Natl Acad Sci U S A. 1988 Jul;85(14):5190-4 stated that "An interaction via the PTK could act to regulate the cooperative interaction between the T3-Ti complex and the CD4 antigen" (last para 6-8 lines from the end). Also Burgess et al Eur J Immunol. 1991 Jul;21(7):1663-8 showed that direct CD4-lck and TCR biochemically.

Reviewer #2 (Remarks to the Author):

This interesting and well-written manuscript from Cheng Zhu's laboratory reports the effect of the PD-1/PD-L1 interaction on the TCR/pMHC interaction as measured using two of their powerful 2D micropipette binding assays.

Their key observation is that this interaction inhibits the TCR/pMHC interaction and reduces its stability. Their mechanistic studies suggest that this is because PD-1 engagement inhibits positive cooperativity between the TCR and CD8. Taken together with previous studies by them and others, this is consistent with a model whereby PD-1 recruits SHP-1/2 which inhibits phosphorylation of CD3 and/or ZAP-70.

This is a novel and important study which convincingly illustrates a key mechanism by which PD-1 works.

This study would be enhanced if the following issues could be addressed.

(1) It is unclear why they have performed the experiments shown in Fig 5 and why this data is at the end of the paper. This shows the effect of PD-1 engagement on signalling downstream of TCR engagement, namely intracellular calcium.

One obvious rationale for this experiment and the experiment shown in Fig 1 (which looks at another downstream response - cell spreading) is to demonstrate that the conditions they use to engage TCR and PD-1 have the expected functional effects. In this case it makes sense to bring Fig 5 and discussion thereof forward, perhaps combining it with Fig 1 or making it a supplementary figure. It would also make sense to verify that way the ligands are presented in these assays is comparable to the methods used in the micropipette binding experiments. Are the glass beads coated with the same densities of pMHC and PD-L1 as the glass coverslips used for microscopy?

(2) When describing the micropipette measurements it is not clear what sequence they were performed in. This is important as the same T cell is used repeatedly to test binding to different surfaces, and binding that triggers the TCR could affect subsequent responses. It is mentioned in the Fig 2 legend that a random sequence was used but this should be clarified. At what 'level' was randomisation done. Between each contact, between each contact time, between each surface?

(3) It is striking that adhesion frequency for the PD-1/PD-L1 combination increases so much

between 0.5 s (Fig. 2d) and 5 s (Fig. 2b) contact durations, especially compared with the TCR/pMHC. This implies that this interaction has slow kinetics. The authors do not comment on this.

(4) It is also striking that PD-1/PD-L1 adhesion frequency at 5 s is higher than that seen for TCR/pMHC (Fig. 2b). This contrasts with the spreading assay where the PD-1/PD-L1 interaction resulted in much lower spreading than the TCR/pMHC interaction. Can the authors comment on this?

(5) It is not clear why the authors performed the experiment in Sup Fig. 1 and what the results demonstrate.

(6) In the supplementary Table 1 they provide information on the number of repeats for each condition for bond life time measurements. However it is unclear what constitutes 1 repeat. Is it one cycle of contact/force clamp/bond break? If so were they all using a single cell or were a number of cells tested for each condition?

REVIEWER COMMENTS

Reviewer #1 (Remarks to the Author):

The paper by Li and coworkers document inhibitory effects of PDL1-PD-1 ligation on CD8-TCR cooperativity. It is an interesting and potentially important study with elegant technology, but there are some issues. A central missing experiment is to analyse effects in which CD8 has been mutated to prevent the binding of Lck. There exist CD8- T-cell lines and most specifically, CD8- T-cells can be isolated from primary T-cells (also CD4-) and transduced or transfected with the mutant. At a minimum, the CD8-Lck binding mutant can be overexpressed in T-cells under conditions where it interferes with responses to gp33 presentation responses due to the absence of Lck binding, could be differentially labelled and assessed for different effects.

We truly appreciate the reviewer's critical suggestion and added new data (Fig. 5 and Supplementary Fig. 3) to test the effect of disrupting CD8 binding to Lck. As suggested, we expressed CD8WT or CD8SKSP mutant (loss of Lck binding) in P14 DN thymocytes and analyzed the bond number using our 2D kinetic analysis. We observed the elimination of both TCR-CD8 positive cooperation (Fig. 5c, d) and TCR--PD-1 negative cooperation (Fig. 5d) by CD8SKSP mutations. Our new data greatly strengthen the conclusion of PD-1 suppression of TCR-CD8 cooperativity.

Specific Points

1) Figure 1: looks convincing but more information on numbers of cells examined is needed as well as control co-ligation with another receptor-ligand pair.

We added cell number for each group and tested the co-ligation using gp33:H2-D^b and ICAM-1. Our new data (Fig. 1f-g) demonstrate an enhancement of pMHC-mediate cell spreading by co-presenting ICAM-1. This is consistent with previous studies of TCR-LFA-1 crosstalk (ref 33 and 34) and further confirms the inhibitory effect of PD-1 is specific.

2) "negative cooperativity" seems an oxymoron and leads to confusion. Cooperativity cannot by definition be negative. You have cooperativity that is inhibited by PDL-1. Does the term need to be changed, perhaps "inhibited cooperativity", "induced dispersion"?

We acknowledge the reviewer's suggestion. However, we would like to keep the term "negative cooperativity" as it is the common term for the phenomenon in which binding of a first ligand decreases subsequent binding.

3) Fig 2: looks convincing. Panel g is unclear PDL1 should be replaced by no peptide PDL-1. Need control with no PDL-1 and no peptide.

We apologize for the confusion. The purpose of panel g-i in Fig. 2 is to test whether the bond number reduction is due to the specific effect of PD-1 or to the physical interference of the two ligands (gp33:H2-D^b and PD-L1) when co-presented. The results suggest the former case. We added negative controls showing that RBC binding to P14 T cells were abolished when no ligands (SA only) were presented or even replacing the cognate peptide gp33 with gp276 on the same H2-D^b MHC (Supplementary Fig. 1). We note that there's no cooperativity to analyze if neither ligand gives binding.

4) How do the authors reconcile their findings with the claim by Gascoigne lab claiming that TCR signalling is triggered by CD8 not associated with Lck (Nat Commun 5, 5624 (2014)).

We do not see conflicts between our model and that of Gascoigne's. We cited this paper (which the Zhu lab contributed) in the discussion of the original manuscript and stated that "Studies disrupting Lck association with CD8/CD4 or inhibiting its activity suggested that the initial step of TCR triggering is likely mediated by Lck not associated with co-receptors^{25,65,66}." This refers to the TCR triggering step as illustrated in Supplementary Fig. 4a. Our model (and also Gascoigne's) suggests an augmentation step by CD8 that amplifies the signal (Supplementary Fig. 4b), and our new data of CD8SKSP mutant (Fig. 5) is also consistent with their study.

5) It is not clear that the T-cells responses are CD8 dependent in terms of eliciting proliferation and the generation of cytolytic T-cell responses.

Plenty of *in vitro* studies have demonstrated the CD8 augmentation of TCR antigen recognition and signaling as well as T cell cytokine production. However, as the reviewer pointed out, it is not clear whether T-cell responses (proliferation and cytolytic activities) are CD8-dependent *in vivo*. It is partly due to the challenges of establishing a faithful model of eliminating CD8 co-receptor function without losing CD8⁺ T cells, as CD8 plays a critical role in CD8⁺ T cell development. T. W. Mak's group has developed a mouse model expressing a CD8 α mutant without cytoplasmic domain, where a small portion of the CD4⁻ CD8⁻ tailless mature T cells were found in periphery (PMID: 8223860). These cells were able to expand upon LCMV infection and had moderately reduced cytotoxicity comparing to WT CD8⁺ T cells from control mice. Yet, it is unclear whether the mature CD8⁻ tailless represent a more responsive population during thymic development, so that they could pass positive selection without a functional CD8. Also, LCMV is considered as a model capable of eliciting strong CD8⁺ T cell responses without the help of additional signals (e.g. CD28 costimulation PMID: 11698427, CD4⁺ T cells PMID: 15286726). It is highly likely that CD8 contributes more to T cell responses to weak antigen epitopes as suggested in the *in vitro* studies (PMID: 12594952). As far as our model is concerned, the mechanism is more focused on the crosstalk of TCR, CD8 and PD-1 at proximal level. The ultimate cellular outcome also depends on the cell type and the immunological context.

6) How do the authors reconcile their model with the current model arguing that PD-1 operates primarily during the effector phase of the T-cell response against antigens, in particular tumour neoantigens?

Their findings may be true but of little real consequence to the real effect of PD-1 during the effector phase of the T-cells response. The authors need to address this issue. Do the mechanisms that they describe operate during the killing phase of CTLs or do they think that the other models of PD-1 modulation of responses are incorrect?

We do not see conflicts of our model with existing models of PD-1 function. We would like to note that as previously reviewed (PMID: 21061197, PMID: 32265932), PD-1 inhibits CD8⁺ T cell responses during both effector and exhaustion phases in response to antigen challenges including bacteria, viruses, as well as tumor neoantigens. A thorough investigation of PD-1 during CD8⁺ T cell responses was reported in a previous study of the Ahmed lab (PMID: 20551512), where we used bone marrow chimeric mice to examine the effects of PD-L1 deficiency in hematopoietic or nonhematopoietic cells during LCMV CL-13 infection. We found that PD-L1 expression on hematopoietic cells inhibited CD8⁺ T cell numbers and function after LCMV CL-13 infection. In contrast, PD-L1 expression on nonhematopoietic cells limited viral clearance and immunopathology in infected tissues. These data indicate that PD-1 functions during

both the activation/priming by APCs and the killing of virus-infected cells. As previously mentioned, our model describes a crosstalk at the proximal signaling level with shared major molecular players in T cells from all these immunological contexts. Therefore, instead of conflicting with existing understanding of PD-1 function, our model provides a potential mechanism complementary to the mostly studied downstream effects of PD-1.

7) Although the SMI: 7-Cyclopentyl-5-(4-phenoxyphenyl)-7H-pyrrolo[2,3-d]pyrimidin-4-ylamine only is advertised as Lck specific, it is unclear whether it might also affect other Src kinases. One must use mutated CD8 without Lck binding (Cys mutants) in order to make their claims rigorous and specifically related to CD8 function.

We have added new data of CD8SKSP mutants that support the role of Lck in our model (see also the response to Comment #1).

8) Do inhibitors of ERK affect the processes in the same way as inhibitors of Lck? It would address whether the Lck requirement is specific at a proximal level or involve general calcium-dependent activation events in T-cells.

Previous studies from our group and others have demonstrated that the TCR-CD8 cooperation is not affected by MEK/ERK inhibitor or Ca^{2+} chelator (ref 24). Instead, it is sensitive to perturbations at the proximal signaling level including 1) Lck kinase inhibitors (ref 24, 26), 2) CD3 ζ ITAM Y->F mutations (ref 26), 3) CD8 mutation that abolishes Lck binding (ref 25, 26), and 4) CD8.4 chimera that enhances Lck binding (ref 26). Our Lck inhibitor data and the new CD8SKSP mutant data further confirm the specific requirement of Lck at a proximal level.

9) One hopes that RBC presentation events are equivalent to DC presentation events...a comment is needed.

We did not state or speculate that these two presentations are equivalent. The main purpose of RBC presentation is to use it as an ultra-soft spring to detect the presence of adhesion at a given contact time with single-bond sensitivity (ref 35). It shares similarities with DC in terms of molecular diffusion on the membrane. But the RBC presentation is different in that 1) only ligands of interests are presented, 2) the interface was forced and controlled instead of being developed as a natural and active process, and 3) the presentation involves repeated brief and intermittent contact cycles rather than a continuous contact.

10) Is not "binding-signaling-binding" just classic "inside out signaling"? Do we really need a new term? We have changed "binding-signaling-binding" into "inside-out signaling" as suggested by the reviewer.

11) Are not the effects of SHP2 consistent with its positive signaling role as described in most non-immunological literature?

We agree with the reviewer that SHP2 has largely a positive signaling role in non-immune cells. For reasons we do not fully understand, SHP2 has been identified as a major phosphatase responsible for PD-1's inhibitory function (ref 4-8). The most direct biochemical evidence was reported in ref 13, where SHP2 was recruited to PD-1 cytoplasmic tail and dephosphorylate several signaling molecules of TCR-CD3 and CD28 axis.

12) Introduction: small point but still important. 1st para, line 4 should reference some of the original

seminal papers on the identification of CD4 and CD8-lck from which all subsequent work originated. For example, Rudd et al (1988) Proc Natl Acad Sci U S A. 1988 Jul;85(14):5190-4 stated that "An interaction via the PTK could act to regulate the cooperative interaction between the T3-Ti complex and the CD4 antigen" (last para 6-8 lines from the end). Also Burgess et al Eur J Immunol. 1991 Jul;21(7):1663-8 showed that direct CD4-lck and TCR biochemically.

We thank the reviewer for pointing out the missing references. We have added the citations in introduction and discussion accordingly.

Reviewer #2 (Remarks to the Author):

This interesting and well-written manuscript from Cheng Zhu's laboratory reports the effect of the PD-1/PD-L1 interaction on the TCR/pMHC interaction as measured using two of their powerful 2D micropipette binding assays.

Their key observation is that this interaction inhibits the TCR/pMHC interaction and reduces its stability. Their mechanistic studies suggest that this is because PD-1 engagement inhibits positive cooperativity between the TCR and CD8. Taken together with previous studies by them and others, this is consistent with a model whereby PD-1 recruits SHP-1/2 which inhibits phosphorylation of CD3 and/or ZAP-70.

This is a novel and important study which convincingly illustrates a key mechanism by which PD-1 works.

This study would be enhanced if the following issues could be addressed.

(1) It is unclear why they have performed the experiments shown in Fig 5 and why this data is at the end of the paper. This shows the effect of PD-1 engagement on signalling downstream of TCR engagement, namely intracellular calcium.

One obvious rationale for this experiment and the experiment shown in Fig 1 (which looks at another downstream response - cell spreading) is to demonstrate that the conditions they use to engage TCR and PD-1 have the expected functional effects. In this case it makes sense to bring Fig 5 and discussion thereof forward, perhaps combining it with Fig 1 or making it a supplementary figure.

We thank the reviewer for suggesting rearranging Fig. 5. However, we note that although both cell spreading and Ca^{2+} require TCR downstream signaling, the cell spreading data also evaluate the stability of the interactions, which is the foundation of the antigen recognition and also the main point of Fig. 1. We then zoom into the "interaction" process in Fig. 2 by performing 2D cooperativity analysis, where the signaling-dependent development of contact interface is replaced by well controlled contact. Both figures focus on the same question – whether PD-1 affects TCR antigen recognition. We evaluated the functional outcome by imaging Ca^{2+} after the mechanistic studies (Fig. 3-5).

It would also make sense to verify that way the ligands are presented in these assays is comparable to the methods used in the micropipette binding experiments. Are the glass beads coated with the same densities of pMHC and PD-L1 as the glass coverslips used for microscopy?

Ligand presentation in these assays were not via the same physical/chemical couplings, and thus difficult to directly compare the absolute densities. However, PD-L1 was kept in excess than pMHC in all these assays.

(2) When describing the micropipette measurements it is not clear what sequence they were performed in. This is important as the same T cell is used repeatedly to test binding to different surfaces, and

binding that triggers the TCR could affect subsequent responses. It is mentioned in the Fig 2 legend that a random sequence was used but this should be clarified. At what 'level' was randomisation done. Between each contact, between each contact time, between each surface?

The randomization is between each RBC (surface). The "memory effect" mentioned by the reviewer was minimized by such randomization and a typical 3-5 min gap that is required to switch RBC. We have revised the figure legend as following to clarify this point "A T cell expressing TCR, CD8 and PD-1 (right) was tested against three RBCs bearing pMHC, PD-L1, or both (mix) in random order (left) to generate three adhesion frequencies (P_a 's), one for each RBC after 30-50 repeated touches with the T cell. The bottom row shows the workflow of using the three P_a 's to determine the bond numbers ($\langle n \rangle$'s) and the differential bond number ($\Delta \langle n \rangle$)." .

(3) It is striking that adhesion frequency for the PD-1/PD-L1 combination increases so much between 0.5 s (Fig. 2d) and 5 s (Fig. 2b) contact durations, especially compared with the TCR/pMHC. This implies that this interaction has slow kinetics. The authors do not comment on this.

We cannot directly compare the P_a 's in these groups as these experiments were tested on different days using T cells and RBCs with different receptor and ligand densities. In two of our previously published studies (ref 10 and 11) we reported a complete characterization of both 3D and 2D kinetics of PD-1 ligand interactions, compared it with TCR-pMHC and B7-1-CD28/CTLA-4 interactions and discussed the implications of the differences in affinity and kinetics. The primary focus of varying contact in this experiment is to reduce the ligand binding and signaling without changing ligand density and thereby to test any possibility of direction physical interference of the ligands.

(4) It is also striking that PD-1/PD-L1 adhesion frequency at 5 s is higher than that seen for TCR/pMHC (Fig. 2b). This contrasts with the spreading assay where the PD-1/PD-L1 interaction resulted in much lower spreading than the TCR/pMHC interaction. Can the authors comment on this?

As stated in response to Comment #1, PD-L1 density was higher than pMHC in these assays. When the T cells were pushed into contact with ligand-coated RBC surfaces, PD-L1 induced more bonds than pMHC as shown in Fig. 2b. However, the cell spreading assay involves a spontaneous development of contact interface by the cells through active processes, which depends on the signaling triggered by the ligand presented on the surface. While it has been well documented that TCR triggers cytoskeleton remodeling and cell spreading/adhesion, PD-1 primarily functions as an inhibitory receptor and lacks the signaling capacity to drive cytoskeleton remodeling required for robust cell spreading/adhesion. Moreover, cell spreading also depends on the stability of the receptor-ligand bonds, especially under mechanical forces. Our bond lifetime vs force data (Fig. 4f) suggest that gp33:H2-D^b bonds are more durable than PD-L1 bonds under force.

(5) It is not clear why the authors performed the experiment in Sup Fig. 1 and what the results demonstrate.

Supplementary Fig. 1 in the original manuscript (Sup Fig. 2 in the revised manuscript) demonstrates that the bond reduction effect of PD-1 is also observed on primary P14 cells activated *in vivo* during LCMV infection, not necessarily limited to *in vitro* activated T cells. We have added the sentence "suggesting PD-1-mediated negative cooperativity for *in vivo* activated T cells during responses to antigen".

(6) In the supplementary Table 1 they provide information on the number of repeats for each condition

for bond life time measurements. However it is unclear what constitutes 1 repeat. Is it one cycle of contact/force clamp/bond break?

One repeat/event represents a valid bond lifetime event observed in many cycles of cell-probe contact.

If so were they all using a single cell or were a number of cells tested for each condition?

The events were pooled from a number of cell-probe pairs.

REVIEWER COMMENTS

Reviewer #1 (Remarks to the Author):

The authors have addressed of the raised issues. The new data showing the effects of co-receptor mutants unable to bind to p56lck has greatly improved the paper.

Reviewer #2 (Remarks to the Author):

I am happy with the revised manuscript.